# Microstructural Evolution and Hardness Responses of 7050 Al Alloy during Processing

**DOI:** 10.3390/ma15165629

**Published:** 2022-08-16

**Authors:** Yuting Zhou, Jie Zhou, Xinrui Xiao, Shishan Li, Mingliang Cui, Peng Zhang, Shuai Long, Jiansheng Zhang

**Affiliations:** 1School of Material Science and Engineering, Chongqing University, Chongqing 400044, China; 2Chongqing Jiepin Technology Co., Ltd., Chongqing 400044, China; 3Deyang Wanhang Die Forging Co., Ltd., China National Erzhong Group Co., Deyang 618013, China; 4College of Metallurgy and Material Engineering, Chongqing University of Science and Technology, Chongqing 401331, China

**Keywords:** 7050 alloy, secondary precipitates, microstructural evolution, hardness

## Abstract

The thermal-mechanical process (TMP) plays an important role in the industrial production of 7050 Al alloys for aircraft components. In this work, the microstructural evolution and influence on hardness of a 7050 alloy ingot, during the process from preheating, deformation and cooling to final heat treatment, have been investigated considering the effect of temperature and the post-deformation cooling path. The results showed that an increasing temperature and decreasing cooling rate during TMP can lead to enhanced hardness of the alloy after heat treatment. Moreover, the variation of recrystallization before and after heat treatment was strongly dependent on the cooling path after deformation. Finally, this study provided a general understanding on the relationship between microstructural evolution and harness of the 7050 alloy.

## 1. Introduction

In the aerospace industry, 7xxx series Al (Al-Zn-Mg-Cu) alloys are widely used due to their lightweight, high strength, high fracture toughness and excellent formability [1,2,3,4]. The industrial production of high-strength 7xxx Al alloys mainly involves two processing stages including the thermal-mechanical process (TMP) and heat treatment (HT). Since 7xxx Al alloys are well known to be heat treatable, a further increase in strength with simultaneously good ductility and fracture toughness can be obtained through specific heat treatment process [5,6,7]. However, in the real industrial processing, the homogeneity of product quality, especially for large aircraft components, are also sensitive to TMP due to their complexity and size effect of their structure. Because various parts of the component are inevitably exposed to inhomogeneous processing conditions due to uneven thickness, which significantly affect the uniformity and reliability of final product. Therefore, to meet increasingly high requirements for manufacturing technology of aircraft components, it is necessary to investigate the effect of TMP parameters on microstructure and mechanical properties evolution.

In recent years, some research has been conducted on the effect of processing parameters on microstructure and mechanical properties of 7xxx Al alloys [8,9,10]. For instance, Liu et al. [11] studied the hot deformation behavior of the 7475 aluminum alloys at deformation temperatures ranging from 573 to 723 K and strain rates ranging from 0.01 to 10 s^−1^. They suggested that the safe-deformation regimes for the alloy could be attributed to the continuous dynamic recrystallization and grain boundary sliding. Khan et al. [12] found that Al-Zn-Mg-Cu alloys are sensitive to different strain rates, and the ultimate strength can be enhanced after the T74 heat treatment under strain rates ranging from 1.0 × 10^−3^ s^−1^ to 5.0 × 10^3^ s^−1^ due to the large number of precipitates. Zhang et al. [13] studied the mechanical behavior and microstructure evolution of Al-Zn-Mg-Cu alloy under various strain rates and temperatures. They provided a significant understanding on the dynamic deformation and damage mechanisms of the alloy considering the temperature effect. However, in most of the previous investigations, the effects of deformation behavior under various TMP parameters on the microstructures and mechanical properties were studied at one specific stage. A general understanding of the effects of possible microstructural phenomena on mechanical properties during the whole processing route (both TMP and HT) is lack of attention, while this understanding is critical in designing optimized processing practices.

In the present work, four conventional temperatures and two typical cooling processes during TMP are selected to simulate inhomogeneous processing conditions probably occurring on the production of large aircraft component. To construct a general relationship between the microstructure and mechanical property of 7050 alloy, the development of microstructure and resultant hardness has been investigated during the process from preheat, hot deformation to final heat treatment. Meanwhile, the effects of temperature and the post-deformation cooling on precipitation and deformation restoration mechanisms have been studied. Moreover, the strengthening mechanism and relationship between the microstructure and hardness during the whole processing route were also discussed.

## 2. Experimental Methods

The employed material is a homogenized 7050 aluminum alloy ingot, whose composition is (in wt.%) Al-6.2Zn-2Mg-2.1Cu-0.11Zr-0.07Fe-0.03Si. The microstructure evolution and resultant hardness during the thermal-mechanical process and heat treatment were analyzed by a series of deformation, microstructure, and hardness tests. Firstly, the 7050 ingot was machined into several cylindrical specimens of φ15 mm × 10 mm. As presented in Figure 1, the initial specimens were heated to the required temperatures (380 °C, 400 °C, 420 °C and 440 °C) and with 300 s holding. Then, isothermal compression tests were performed at a constant strain rate of 0.1 s^−1^ to a fixed strain of 0.9, supplied by a Gleeble-3500 machine. After that, the specimens were immediately uninstalled and cooled. Two cooling treatments after deformation, water quenching (30 °C~100 °C/s) and air cooling (30 °C~2 °C/s) were selected. The corresponding samples are individually named ‘DW’ or ‘DA’. Subsequently, these samples were further heat treated in T74 process, where they were heated up to solution temperature, 474 °C, for 1 h, followed by rapidly quenching in warm water (65 °C), and the subsequent two-step aged, at 121 °C, for 6.5 h and 177 °C for 9.5 h. In the same way, the corresponding heat-treated samples are named ‘DWHT’ and ‘DAHT’, respectively.

The samples preserved at the end of each processing step, were wire-electrode cut along central plane for metallographic observation and hardness test. The microstructure morphology, including grain structure and secondary-phase particles, was analyzed using a scanning electron microscope (SEM) equipped with a back-scattered electron signal (BSE), an electron back-scattered diffraction (EBSD) probe, and a transmission electron microscope (TEM). SEM and EBSD samples were electro-polished in a solution of 30 vol% HNO3 and 70 vol% methanol below −30 °C at 20 V. Thin foils (~50 nm in thickness) for TEM analysis were obtained by twin-jet electro-polishing. SEM and TEM allow for easy observation of second-phase particles. EBSD mappings allow for easy identification of different grain structures. The step size utilized for EBSD characterization is 0.5 μm. Finally, Vickers hardness was measured at a standard method with 500 gf in load for 10 s holding.

## 3. Results and Discussion

### 3.1. Changes in Hardness

The variations of Vickers hardness of the 7050 alloy during the processes from preheating, deformation and cooling, to final heat treatment were measured and presented in Figure 2. All data were measured after the material cooled. As expected, the hardness bore an upward trend. The initial hardness value of as-received materials was 98 HV, which increased to 125 HV, 135 HV, 142 HV and 142 HV after preheating to 380 °C, 400 °C, 420 °C and 440 °C for 300 s. After the isothermal compression, the hardness value of the material which was fast cooled in room-temperature water (DW) followed an abrupt drop. However, a slight rise was observed in the material slow cooled in the air (DA). According to the results, the hardness value of fast-cooled material (DW) decreased to 115 HV, 121 HV, 133 HV and 134 HV, which was 8%, 10.8%, 6.3% and 5.6% lower than the preheated, respectively. The hardness of the DA material increased to 134 HV, 132 HV, 140 HV and 149 HV, which was 16.5%, 9.1%, 5.3% and 11.2% higher than that of the DW material, respectively. Subsequently, all values of hardness increased significantly after heat treatment. The DAHT materials exhibited a more excellent hardness value than the DWHT materials. In Figure 2, the hardness values of the DWHT materials are 150 HV, 160 HV, 161 HV and 157 HV, respectively, while they increased to 164 HV, 166 HV, 174 HV and 178 HV in the DAHT materials, representing an increase of 9.3%, 3.8%, 8.1% and 13.4%, respectively, versus the DWHT material. It can be found that the higher the preheating temperature, the larger the hardness obtained in each process. Moreover, the longer cooling duration after deformation is beneficial to the hardness performance.

### 3.2. Effects of Precipitates on Hardness

#### 3.2.1. Precipitate Characteristics

The initial microstructure (as shown in Figure 3) consists of an equiaxed grain about ~200 μm and contains a variety of secondary-phase particles. These particles are inhomogeneously distributed in grains interior or continuously located along grain boundaries. Among them (Figure 3b), coarse rod-like particles are about ~1–13 μm in length and ~0.5–1.5 μm in thickness. Fine platelet-like or spherical particles are ~1 μm in length. According to the previous studies [14], the coarse particles within grains are probably to be η-phase (MgZn_2_), S-phase (Al_2_CuMg) and T-phase (Al_2_Zn_3_Mg_3_), etc. The continuous chain-like arrangement of particles along grain boundaries is likely to be Mg(Zn_2_, AlCu). Large intermetallics aggregating at intersections of grain boundaries maybe insoluble, which are enriched with Mg, Cu and Fe (as shown the EDS results in Figure 3a).

Figure 4 shows the secondary particles of the alloy after preheating, where the distribution and geometrical specification has become highly different. It can be seen that the constituent particles gradually dissolved into the Al matrix with increase in temperature. From the observation, some pre-existing fine platelet-like or spherical particles has dissolved at 380 °C. Subsequently, coarse rod-like particles gradually became small and granular at elevated temperatures, along with noticeably reduction in number density of precipitates. Furthermore, Figure 5 shows deformed microstructures with regard to different preheating temperature. Figure 5a–d correspond to the DW material. Figure 5e–f correspond to the DA material. It can be seen that the initial equiaxed grains became elongated under isothermal compression. Grain-boundary intermetallics are fragmented to form spherical and irregular morphologies from the original morphology before compression. Meanwhile, there is a slight variation in the precipitate morphology before and after compression. Yet, few appreciable differences can be observed between DW and DA microstructures at each temperature.

However, according to the TEM images in Figure 6, the post-deformation cooling process has introduced a remarkable difference in the morphology of ageing precipitates. As shown in the bright field (BF) images of TEM, the DWHT and DAHT microstructures both contained densely distributed ultrafine features with sizes of ~2–24 nm. There are fine spherical precipitates with diameters of about ~5–8 nm, which are dominant in the DAHT sample. However, fine platelets about ~8–15 nm in length and ~2–5 nm in thicknesses are only observed in the DWHT sample. Coarser rod-like precipitates with lengths of ~20–30 nm and with thicknesses of ~4–14 nm and spherical particles with diameters of ~24 nm are observed in both microstructures. Among them, spherical ones belong to the Al_3_Zr dispersoids [15].

#### 3.2.2. Statistical Analysis of Matrix Precipitates

It is well accepted that the size parameters of the precipitates have a strong influence on mechanical strength of 7050 alloys. Therefore, to reveal the evolution of precipitates during TMP and HT, we calculated the average precipitates size of the preheated, deformed and heat-treated samples, respectively, using the methodology proposed in Ref. [16]. The characteristic size is an average radius (*r_i_*) between the measured long and short axes of the matrix precipitate, as the following equation:(1)$$r_{i}=\left(D+T\right)/2$$
where *T* is the thickness of the observed round particles or edge-on platelets, and *D* is diameter of the round particles or edge-on platelets. Standard deviation (*σ*) is determined according to the following equation:(2)$$\sigma =\sqrt{\frac{1}{N}\overset{N}{\underset{i=1}{\sum }}{\left(r_{i}-r_{mean}\right)}^{2}}$$
where *N* is the number of the measured precipitates, *r_i_* is the measured radius, and *r_mean_* is the mean radius of the measured precipitates.

In addition, the precipitate volume fraction of the samples under different temperatures can be calculated by [17]:(3)$$V_{sphere}=(4\pi /3)R_{mean}{}^{3}$$
(4)$$V_{platelet}=(\pi /4)TD^{2}$$
(5)$$V_{cor}=(4\pi /3)R_{mean}{}^{3}\cdot N_{platelets}$$
(6)$$V_{p}=V_{sphere}+2V_{platelet}-V_{cor}$$
where *V_sphere_* represents the volume fraction of observed round particles, *V_sphere_* represents the volume fraction of observed edge-on platelets, *V_cor_* represents the volume fraction of observed round precipitates which actually are platelets, *R_mean_* is the average of the precipitates and *N_platelets_* is the density of edge-on platelets.

The resulting distributions of the precipitate size in the preheated samples (randomly selected in Figure 4 using a yellow rectangular box) are depicted in Figure 7. It can be seen that the radii of most precipitates in the sample exposed at 380 °C (Figure 7a) are smaller than 2 μm with ~20.1% of precipitates in the range of 2–5.3 μm. As a result, the calculated average precipitate radius is 1.27 ± 0.97 μm. As the temperature increases up to 400 °C (Figure 7b), the precipitate radius decreases slightly, with a size interval of 2–4.4 μm occupying 12.2% of the whole. Hence, the average precipitate size is 1.02 ± 0.79 μm. In the sample after thermal exposure at 420 °C (Figure 7c), many precipitates with granular shape and a size larger than 2 μm only occupy less than 4.9%. The average precipitate radius reaches 1.19 ± 0.57 μm. Similarly, in the sample under temperature of 440 °C, the radii of precipitates with interval size of 1–2 μm reaches almost 92.2% of the whole. Moreover, the average precipitate size is 1.18 ± 0.63 μm. According to the statistical results, it can be confirmed that the dissolution was firstly occurred in the precipitates with a radius below 1 μm, then the number density of the precipitates in the range 2–5.5 μm reduced significantly with the increase in preheating temperature. The resulting precipitate volume fractions of the preheated samples (as presented in Table 1) at 380 °C, 400 °C, 420 °C, and 440 °C are 2.70%, 2.02%, 0.59% and 0.48%, respectively. It can be concluded that the precipitate density decreased continuously with the increase in preheating temperature until the temperature rose to 420 °C.

The size distributions of precipitates in the DW samples (randomly selected in Figure 5 using a white rectangular box) are presented in Figure 8. It can be seen that the pre-existing precipitates are slightly larger than that of their initial counterparts before compression. In the sample deformed at 380 °C, the radii of precipitates with a size interval of 2–5.2 μm occupies 21.2% of the whole. The resulting average size is 1.30 ± 0.94 μm. At 400 °C, the precipitate radius with a size larger than 2 μm reaches ~17.8%. The corresponding average precipitate size is 1.44 ± 0.76 μm. As for the samples deformed at 420 °C, many precipitates are still with granular morphology with a size interval of 0–2 μm occupying 89.6% of the whole. Moreover, the average precipitate radius reaches 1.24 ± 0.47 μm. With temperature increases up to 440 °C, the radii of precipitates smaller than 2 μm occupies more than 95% with relatively small average radius of 1.02 ± 0.35 μm. The resultant volume fractions of precipitates under each temperature are 5.88%, 6.52%, 3.50% and 1.72%, respectively, which are significantly higher than the samples before deformation. It can be concluded that dynamic precipitation may have occurred in the interspaces of pre-existing precipitates during plastic straining, according to Refs. [18,19].

In addition, the precipitate size distribution of DWHT and DAHT samples are given in Figure 9. It can be seen that the radii of most precipitates in the DWHT sample at condition of 420 °C (Figure 9a) are smaller than 8 nm, with ~26% of precipitates in the range of 8–14 nm and ~10% precipitates in the range of 14–24 nm. As a result, the calculated average precipitate radius is 7.97 ± 3.79 nm. In the DAHT sample (Figure 9b), more precipitates with larger size and the size interval of 8–14 nm reaches almost 40% of the whole, and those of the size over 14 nm are ~11%. The average precipitate size reaches 9.25 ± 3.88 nm. The resulting precipitate volume fractions of the DWHT and DAHT samples (as presented in Table 1) are 9.94% and 21.6%, respectively. It suggests that slow cooling rate after deformation can promotes the growth of ageing precipitates with standard T74 heat treatment process.

#### 3.2.3. Effects of Precipitates on Hardness

Since 7050 alloy is one kind of heat-treatable alloy, the distribution and geometrical specifications of secondary precipitates can strongly affect the mechanical property of the alloy. There are two kinds of “shearable” and “non-shearable” particles. The “shearable” particles can be sheared by moving dislocations and then flow with the matrix. Their contribution to the strength of the alloy involves the shear resistance of one particle, their population, and the flexibility of the dislocation with which they interact, attributed to dispersoid strengthening. It can be described as following equation [20]:(7)$$∆\sigma_{1}=c_{1}f^{m}r^{n}$$
where $c_{1}$, *m* and *n* are constants, *f* is the volume fraction of precipitates, *r* is the average precipitate radius, and B_1_ = $r^{n}$; for most dislocation-particle interactions, both *m* and *n* have the value of 1/2. Obviously, Equation (7) represents the higher the volume fraction and size of the “shearable” particles, the better the strengthening effectiveness. As known, most dispersed ageing precipitates (GP zone and η’) in Figure 8 could be considered “shearable” [7]. Therefore, consistent with Equation (7), the DAHT sample decorated much more precipitates has greater hardness than the DWHT samples.

On the other hand, coarse particles primarily remaining before heat treatment (in Figure 4 and Figure 5) could belong to “non-shearable” particles. The dislocations could only bypass but not cut them, leading to dislocation loops around them. Each dislocation loop could act as reverse stress on the dislocation source so that the resistance of movement of dislocation is considerable, causing the increase in strength and reduction in ductility. The strengthening effect is hence transformed into precipitation strengthening. It has been established as follows [20]:(8)$$∆\sigma_{2}=C_{2}f^{1/2}r^{-1}$$
where $C_{2}$ is constant, and B_2_ = $f^{1/2}r^{-1}$. According to Equation (8), in the material dominated by “non-shearable” particles, reducing precipitate size (with constant volume fraction) or increasing precipitate volume fraction leads to the increased strength. According to the calculated B_2_ values (in Table 1), the hardness value of preheated or deformed materials should decrease with rising temperature. However, considering an opposite result obtained in Figure 3, it indicates that not only precipitation strengthening operates in those cooled materials, but also additional strengthening mechanism must occur and dominate the effectiveness of strengthening. Throughout the variations of hardness during TMP and HT, higher temperature and slower cooling rate are always correlated to greater hardness. Hence, it can be concluded that the additional strengthening effects should come from dispersion precipitation during the cooling process. Since increasing the starting cooling temperature and decreasing the cooling rate leads to a long period of cooling time, more dispersoids tend to precipitate from the Al matrix, thereby enhancing the hardness of the alloy. Combined with the performance in the heat-treated materials, it suggests that the post-deformation cooling, with a typical industrial rate, also has a markable influence on the ageing response of the 7050 alloy.

### 3.3. Effects of Microstructure on Hardness

#### 3.3.1. Microstructures before Heat Treatment

Dynamic recovery (DRV) is the dominant deformation mechanism of Al alloys, which is one kind of alloy with high stacking fault energy. Figure 10 and Figure 11 show the typical DRV microstructure. Due to the frequent operation of DRV at elevated temperature, dislocation density decreases with the formation of subgrains. Nevertheless, deformation bands (DBs) are observed along the deformation direction at a relatively low temperature of 380 °C, as several low-grain boundaries (LABs) traversed and intersected in the interior of the original grains.

According to the EBSD results, some recrystallized grains can be also obtained in DW and DA microstructures. As shown in Figure 12, the observed recrystallized grains can be divided into three categories: (a) most nuclei had formed HABs but still with partial LABs; (b) some nuclei were incompletely formed, still containing many dislocations inside; (c) other nuclei were completely formed with new HABs and free dislocation. Obviously, the first two nuclei tends to occur in the highly strained regions containing many substructures, where LABs are progressively converted into HABs, thereby forming fine new grains. Such nucleation of new grains is only related to the dislocation glide and climb at the expense of the surrounding regions, referring to continuous dynamic recrystallization (CDRX) [21]. The size of third grains is much smaller than the other two. This recrystallization nuclei is prone to occur in the regions with low stored energy, either by a less dense network of dislocations or lower misorientation gradients. It grows up by consuming neighboring grains with high stored energy to form strain-free grains. The nucleation characteristic is consistent with the mechanism of strain-induced boundary migration (SIBM). As reported in Ref. [22], SIBM is one kind of discontinuous dynamic recrystallization (DDRX), which rarely occurs in Al alloy. The resulting volume fraction of recrystallization has been presented in Figure 15. It appears that the recrystallization fraction in the DW samples is relatively small, ranging from 3.83% to 5.33%, while that in the DA sample is 9.7%, 8.95%, 15.2% and 16.6%, respectively.

#### 3.3.2. Microstructures after Heat Treatment

In the DWHT sample at condition of 380 °C (Figure 13a,b), some recrystallized grains are enormous and interconnected, forming recrystallization bands aligned along original high-angle boundaries. As the temperature increases (Figure 13c–h), the number of such abnormal recrystallized grains declined significantly. They display as isolated patches of the matrix. Generally, abnormal growth of recrystallized grains tends to occur in those who grow up by preferentially consuming their neighbors. Coarse constituent particles with sizes of >1 μm have been considered as one typical origin for such growth advantage. It is easy to understand that the abnormal recrystallization in the DWHT samples was accelerated at the solution treatment. In this period, the pre-existing particles gradually dissolved into the Al matrix to form a supersaturated solid solution, weakening their pinning effect on grain boundaries. Therefore, grain boundary migration was promoted until they contacted each other to form a recrystallization band. The slow cooling rate corresponding to the DAHT samples in Figure 14 allows the occurrence of static restoration after deformation, and then the elimination of stress concentration and relaxation of deformed microstructure, thereby reducing stored energy/driving force in the system. It is while high stored energy remaining in the DW system before heat treatment promoted the occurrence of abnormal recrystallization.

After heat treatment, the volume fraction of recrystallized grains increased significantly as shown in Figure 15. Meanwhile, the recrystallization fraction of the DAHT samples varies from 16.8% to 20%. In the DWHT samples, the volume fraction of recrystallization bears a downward trend, decreasing from 43.8% at 380 °C to 7.31% at 440 °C. It should be noted that the increased recrystallization regions are somewhat responsible for the softening effect. At first, recrystallized grains are well known to be softer than substructured ones. Secondly, for 7050 alloys, coherent–incoherent conversion of Al_3_Zr affected by recrystallization will promote the nucleation of coarse precipitates on incoherent Al_3_Zr dispersoids during the quench after solution. It drains alloying elements from the matrix so that there is low solute to form hardening precipitates during ageing, thereby causing the reduced effect of dispersoid strengthening. However, otherwise, the substructure of the recovered state can provide dispersed nucleation sites for η’ and η hardening precipitates, benefiting ageing responses. Therefore, from the above perspective about grain structure, the DAHT samples should have larger hardness values than the DWHT samples.

### 3.4. Strengthening Mechanisms and Hardness Variations

Considering the nature of precipitation-strengthened 7xxx series Al alloys, the following strengthening mechanisms need to be used to explain the variations of hardness during TMP and HT: grain-boundary strengthening, solid-solution strengthening, dislocation strengthening and precipitate/dispersoid strengthening.

There is no doubt that the preheating process of the 7050 ingot involves a sequence of precipitation. Most research works have reported similar precipitation events for the heating stage of 7xxx series Al alloys [23,24]. Generally, at the early stage of the heating procedure, the fine constituent particles firstly dissolve into the Al matrix along with disappearance of composition segregation. The material can hence be strengthened due to the homogenization and redistribution of solute atoms in the matrix. As the heating procedure proceeds, new fine precipitates (η’ and η) start form from the primary supersaturated solid solution and dispersed throughout the matrix. This causes a significant increase in hardness, attributed to dispersoid strengthening. In the present paper, the material was further heated to 380 °C–440 °C. It is a conventional temperature span for hot deformation of 7050 alloys. In this span, pre-existing η-phase particles rapidly dissolve and become granular, due to the increased solubility of alloying elements at elevated temperature. As a result, the effect of solid-solution strengthening is enhanced with increased residual solute concentrations of Zn, Mg and Cu in the Al matrix [25]. In addition, since the volume fraction of matrix precipitates remarkably decreased from 2.7% to 0.48%, the precipitate strengthening effect is significantly weakened (as described in Equation (2)). Hence, the rising trend of hardness should become slow, or even turn to a downward trend.

In our work, all hardness values were measured after the material cooled. As known, during the cooling procedure, the total precipitation kinetics slows down, so that fine precipitates re-form in the interspaces of pre-existing precipitates. These newly formed phases may slightly coarsen at the terminal stage of the cooling procedure, but the number density is much higher than that before cooling [26,27]. Accordingly, dispersoid strengthening leads to an appreciable increase in hardness. Higher starting cooling stage temperature and slower cooling rate corresponds to long-time exposure for the alloy to elevated temperature, allowing much hardening particles to precipitate, thereby causing a larger hardness value. This is consistent with the test results in Figure 3. Therefore, it can be concluded that dispersoid strengthening on the cooling process dominates the hardness variation in this paper.

One of the most significant consequences of hot deformation for 7050 alloys is grain size refinement, creating a high density of dislocations and (sub)grain boundaries. They impede dislocation movement and dislocation propagation to adjacent grains, thereby strengthening the materials, referring to dislocation/(sub)grain boundary strengthening [25,28]. From Figure 15, the strengthening effect of dislocations on deformed materials decreases with an increasing temperature. While the fraction of LABs does not change significantly with temperature, so (sub)grain boundary strengthening can only provide a relatively small contribution to the difference in hardness between each temperature. On the other hand, plastic strain can induce a significant acceleration of the precipitate coarsening processes [18]. As a result, as shown in Table 1, the average size and volume fraction of matrix particles of the deformed materials increased significantly, compared with the preheated materials. Meanwhile, the hardness of DW materials decreases by 5.6–10.8%. Herein, it is believed that dynamic precipitation during plastic deformation reduces the degree of supersaturation, slowing down the nucleation rate of precipitates during the cooling process, thereby reducing the hardness [29,30].

In this paper, the solution treatment operates at a temperature of 474 °C for 1 h. Under this condition, η-MgZn_2_ particles will completely dissolve into the matrix [31,32]. Hence, the hardness will follow a rapidly descending trend due to the formation of supersaturated solid solution. Subsequently, the hardness will bear a sharp up-trend as a consequence of strong dispersoid strengthening during ageing. It is worthy to note that the hardness of heat-treated materials increases with an increasing temperature and decreasing post-deformation cooling rate. It indicates the important effects of TMP in industrial production.

For the sake of explanation of the relationship between microstructural conditions and resultant hardness of the 7050 alloy during TMP and HT, we characterized three temperatures: critical temperature (CT~120 °C) [2], dissolution temperature (DT~300 °C) [6], and solution temperature (ST~470 °C). It is reasonable to assume that when the temperature exceeds the CT, new fine precipitates would progressively precipitate in the matrix and slightly coarsen with an increasing temperature. In this period, the hardness increases rapidly. As the temperature rises to DT, the pre-existing particles would start to dissolve into the matrix. It causes a slight decrease in hardness, due to the reduced volume fraction of precipitates. At the ST, the constituent particles will completely dissolve into the matrix, where solid-solution strengthening dominates the variation of hardness, causing a remarkable decrease in hardness. Finally, a general interpretation of the effects of possible microstructural phenomena on hardness of the 7050 alloy during the whole processing route is given in Figure 16.

## 4. Conclusions

A study has been performed on the effect of temperature and the post-deformation cooling path on the microstructural evolution of a 7050 Al alloy during the thermal-mechanical process and heat treatment. The strengthening mechanism and relationship between microstructures and hardness during the whole processing route have also been investigated. The following conclusions may be drawn from this work:Continuous dissolution during preheat was firstly occurred in the pre-existing precipitates with a radius below 1 μm, then in the precipitates with a size interval of 2~5.5 μm with an increasing temperature from 380 °C to 440 °C. The resulting volume fractions of the matrix precipitate decreased from 2.70% to 0.48%. In addition, dynamic precipitation may have occurred in the interspaces of pre-existing precipitates during plastic deformation, causing a remarkable increase in precipitates volume fraction in the range of 1.72–5.88%.Densely distributed ultrafine features were both obtained in the heat-treated materials. At condition of 420 °C, fine spherical precipitates with diameters of about ~5–8 nm, which were dominant in the DAHT sample. However, fine platelets about ~8–15 nm in lengths and ~2–5 nm in thicknesses were only observed in the DWHT sample. The resulting precipitate volume fractions of the DWHT and DAHT samples were 9.94% and 21.6%, respectively.The variation of recrystallization fraction before and after heat treatment was strongly dependent on the post-deformation cooling path. Higher volume fraction of recrystallized grains was usually obtained in the material slow cooled after deformation. Exceptionally, extremely high recrystallization fraction of 43.8% was obtained in the DWHT sample corresponding to 380 °C, due to the occurrence of abnormal recrystallization.The hardness bore an upward trend during the processes from preheating, deformation and cooling, to final heat treatment. In addition, dispersoid strengthening dominated the hardness variations on the cooling process. As a result, increasing the starting cooling temperature and decreasing the cooling rate allowed more dispersoids to precipitate, thereby enhancing the hardness of the alloy.

## Figures and Tables

**Figure 1 materials-15-05629-f001:**
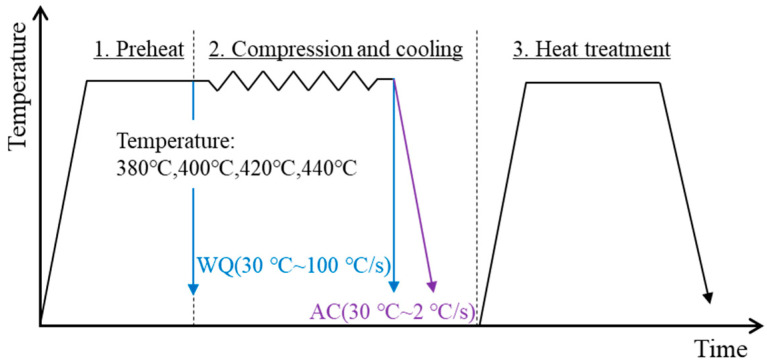
Experimental procedure of the as-received 7050 Al alloy.

**Figure 2 materials-15-05629-f002:**
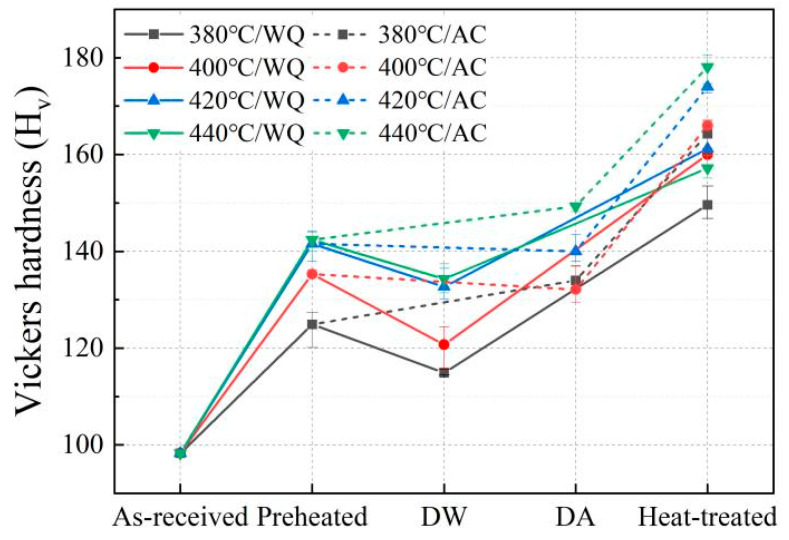
Hardness changes in the 7050 alloy during the whole processing route.

**Figure 3 materials-15-05629-f003:**
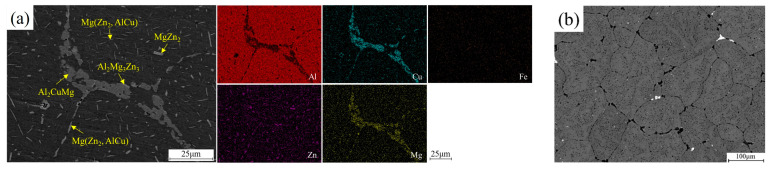
(**a**) EDS results and (**b**) SEM image of the as-received 7050 Al alloy.

**Figure 4 materials-15-05629-f004:**
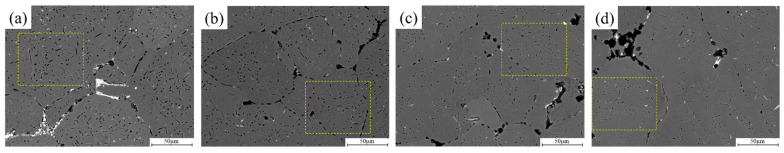
SEM images of the 7050 alloy after preheating to (**a**) 380 °C, (**b**) 400 °C, (**c**) 420 °C, and (**d**) 440 °C, respectively.

**Figure 5 materials-15-05629-f005:**
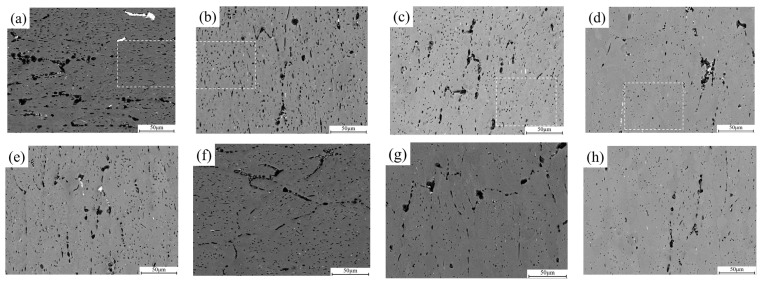
SEM images of the 7050 alloy after deformation and cooling treatment: (**a**,**e**) 380 °C, (**b**,**f**) 400 °C, (**c**,**g**) 420 °C, (**d**,**h**) 440 °C. (**a**–**d**) Correspond to fast quenching in water, and (**e**–**h**) correspond to slow cooling in air.

**Figure 6 materials-15-05629-f006:**
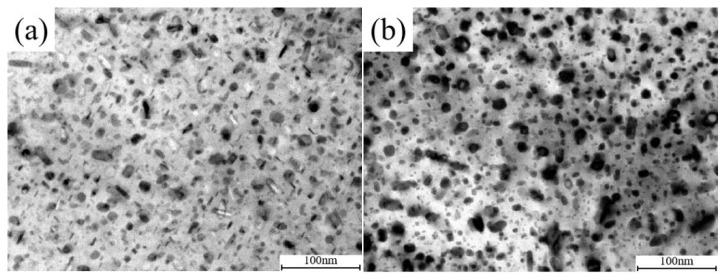
Bright field (BF) images of precipitates morphology of the 7050 alloy after heat treatment corresponding 420 °C: (**a**) represent to the DWHT materials and (**b**) represent to the DAHT materials.

**Figure 7 materials-15-05629-f007:**
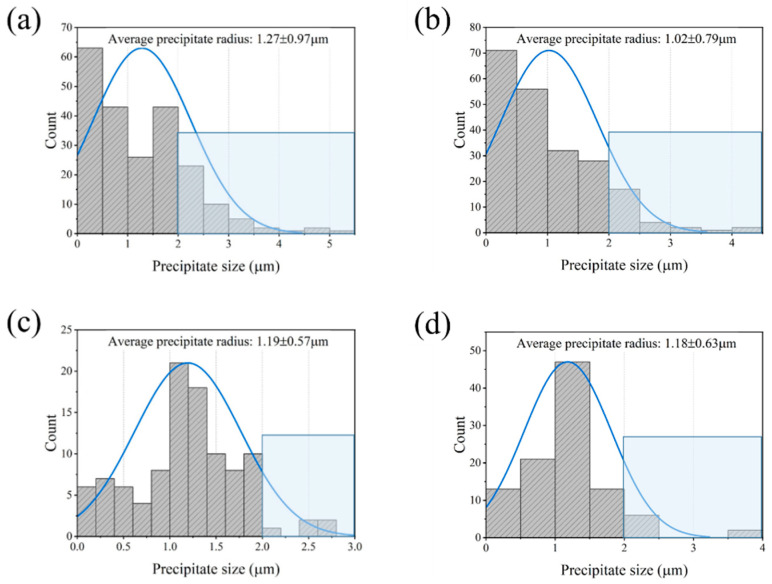
Matrix precipitate size distribution of the 7050 alloy after preheating to (**a**) 380 °C, (**b**) 400 °C, (**c**) 420 °C, and (**d**) 440 °C, respectively.

**Figure 8 materials-15-05629-f008:**
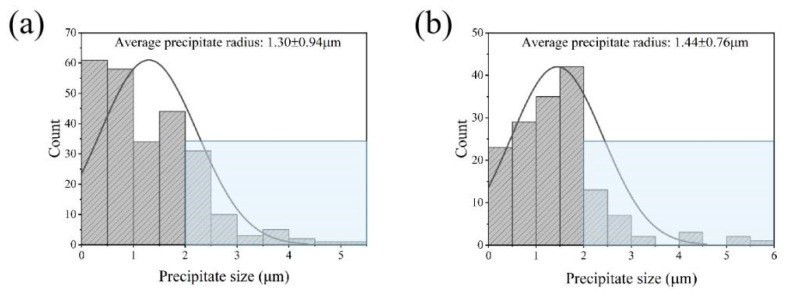
Matrix precipitate size distribution of the DW samples corresponding (**a**) 380 °C, (**b**) 400 °C, (**c**) 420 °C, and (**d**) 440 °C, respectively.

**Figure 9 materials-15-05629-f009:**
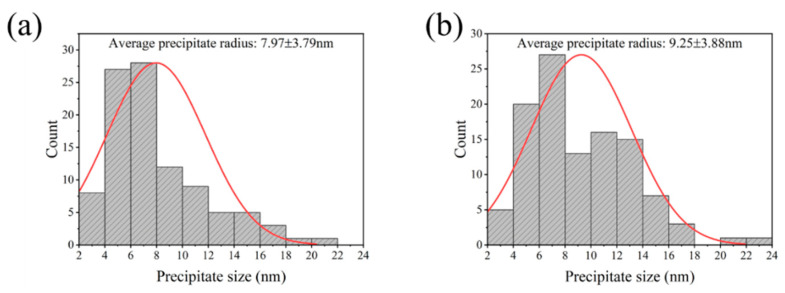
The distribution of precipitate size in the (**a**) DWHT sample and (**b**) DAHT sample at condition of 420 °C.

**Figure 10 materials-15-05629-f010:**
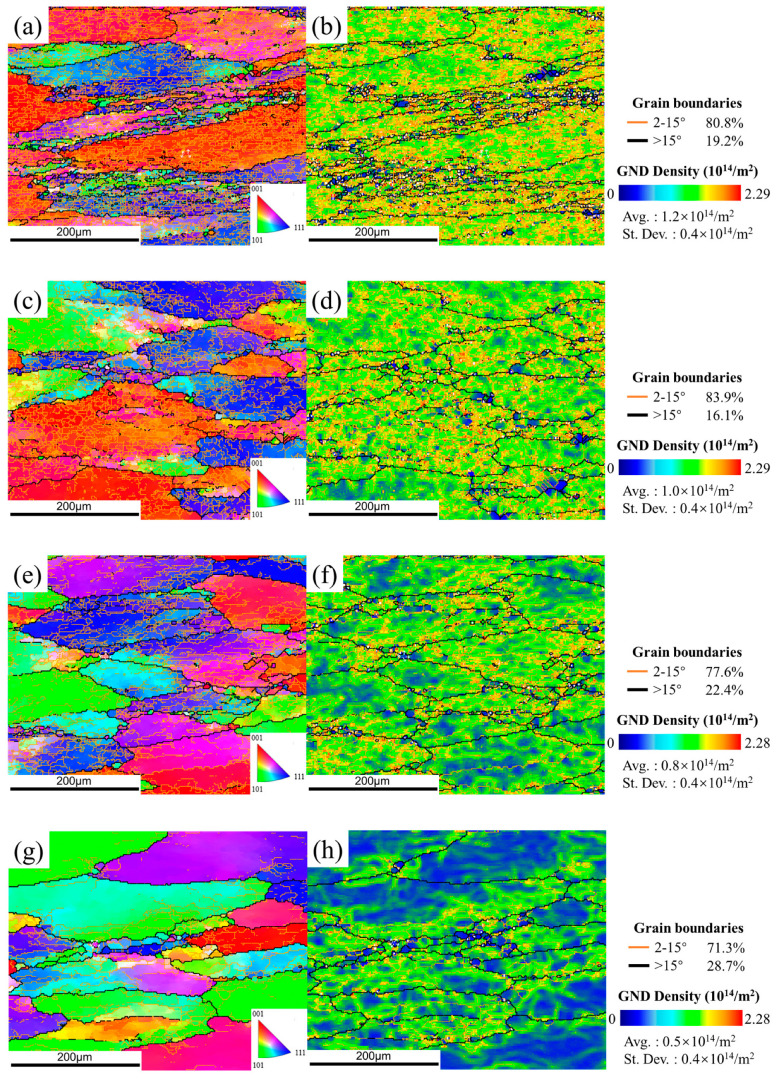
The IPF maps (**a**,**c**,**e**,**g**) and the GND maps (**b**,**d**,**f**,**h**) of the DW samples at conditions of (**a**,**b**) 380 °C, (**c**,**d**) 400 °C, (**e**,**f**) 420 °C, and (**g**,**h**) 440 °C. In the EBSD maps, bold black lines represent the high-angle boundaries (HABs, with misorientation angle >15°) and thin color lines represent low-angle boundaries (LABs, with misorientation angle 2~15°). The color bar of GND images represents the extent of dislocation density or strain in grains: the blue color means the lowest dislocation density/strain areas, the green color means the higher dislocation density/strain areas, and the orange color means the highest dislocation density/strain areas.

**Figure 11 materials-15-05629-f011:**
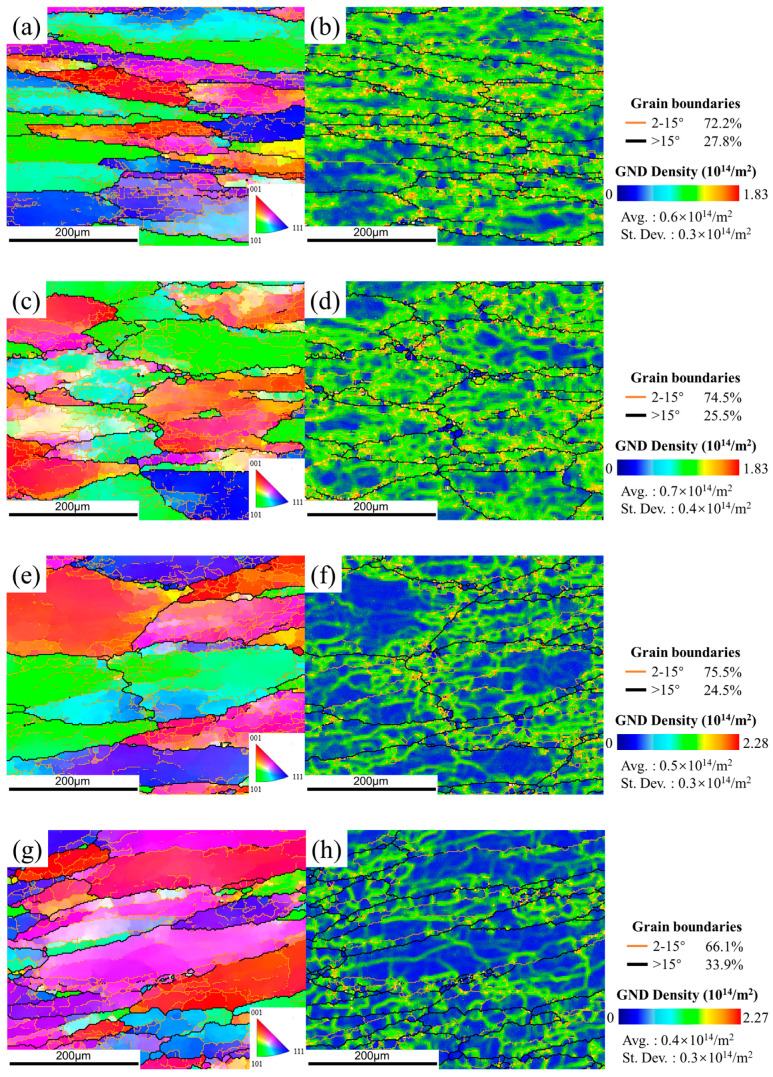
The IPF maps (**a**,**c**,**e**,**g**) and the GND maps (**b**,**d**,**f**,**h**) of the DA samples at conditions of (**a**,**b**) 380 °C, (**c**,**d**) 400 °C, (**e**,**f**) 420 °C, (**g**,**h**) 440 °C.

**Figure 12 materials-15-05629-f012:**
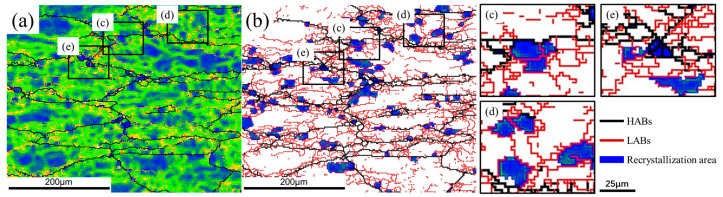
(**a**) The GND map and (**b**) the recrystallized grains of the DA sample corresponding 400 °C. Three types of recrystallization can be obtained: the nuclei (**c**) formed HABs but still with partial LABs; (**d**) incompletely formed inside the initial grains; (**e**) completely formed with new HABs.

**Figure 13 materials-15-05629-f013:**
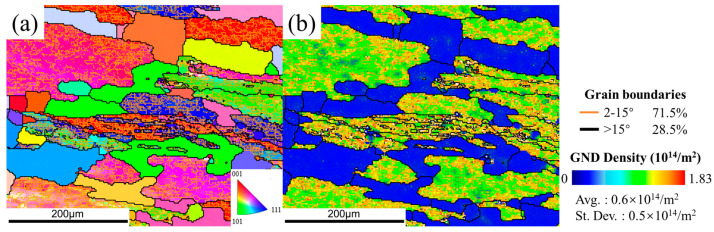
The IPF maps (**a**,**c**,**e**,**g**) and the GND maps (**b**,**d**,**f**,**h**) of the DWHT samples at conditions of (**a**,**b**) 380 °C, (**c**,**d**) 400 °C, (**e**,**f**) 420 °C, and (**g**,**h**) 440 °C.

**Figure 14 materials-15-05629-f014:**
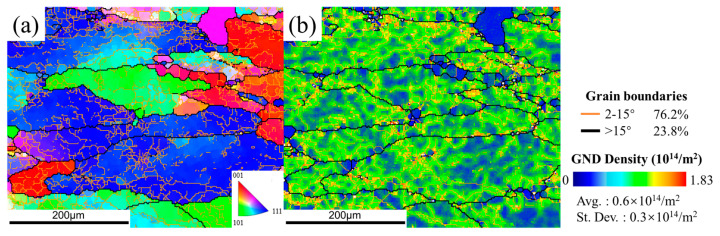
The IPF maps (**a**,**c**,**e**,**g**) and the GND maps (**b**,**d**,**f**,**h**) of the DAHT samples at conditions of (**a**,**b**) 380 °C, (**c**,**d**) 400 °C, (**e**,**f**) 420 °C, and (**g**,**h**) 440 °C.

**Figure 15 materials-15-05629-f015:**
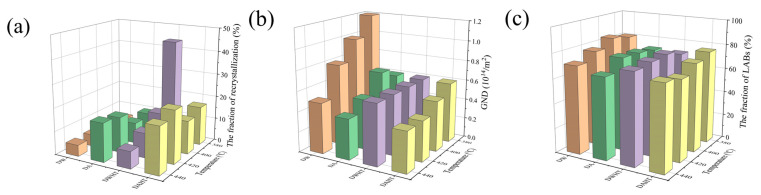
Statistical results of (**a**) the volume fraction of recrystallized grains, (**b**) the GND density, and (**c**) the fraction of LABs at different conditions. The orange histograms correspond to the DW samples; the green histograms correspond to the DA samples; the purple histograms correspond to the DWHT samples; the yellow histograms correspond to the DWHT samples.

**Figure 16 materials-15-05629-f016:**
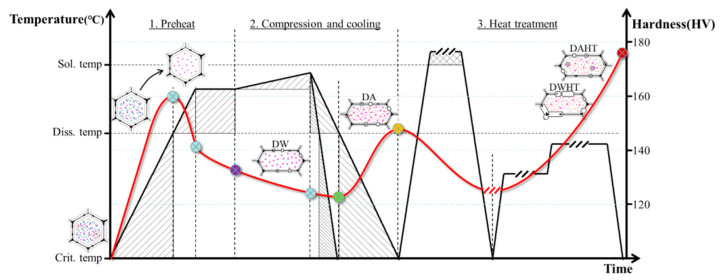
Schematic diagram of the hardness evolution with the possible microstructural phenomena of the 7050 alloy during the whole processing route. The red line corresponds to the changes of hardness.

**Table 1 materials-15-05629-t001:** Statistical results of the precipitates remaining before and after heat treatment.

	Average Precipitate Radius (r_avg_)	The Volume Fraction (V_f_)	$B_{2}(f^{1/2}r^{-1})$	$B_{1}(f^{m}r^{n})$
Preheat-380 °C	1.27 μm	2.70%	1.2841	
Preheat-400 °C	1.02 μm	2.02%	1.3884	
Preheat-420 °C	1.19 μm	0.59%	0.6436	
Preheat-440 °C	1.18 μm	0.48%	0.5865	
DW-380 °C	1.29 μm	5.88%	1.8705	
DW-400 °C	1.44 μm	6.52%	1.7701	
DW-420 °C	1.24 μm	3.50%	1.5071	
DW-440 °C	1.02 μm	1.72%	1.2837	
DWHT	7.96 nm	9.94%		1.1171
DAHT	9.25 nm	21.60%		1.5285

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
