# Peer review of "Microstructural Evolution and Hardness Responses of 7050 Al Alloy during Processing"

_materials, 2022, doi:10.3390/ma15165629_

Round 1
Reviewer 1 Report
In this manuscript, the authors investigated the effects of heat treatments on the microstructural and the hardness evolutions. The topic of the paper is suitable for Materials. Some revisions are to be made before the acceptance of the paper:
- Some comments on how the current results could help understanding the heat treatment selection and the effects on resultant mechanical performances should be added in the end of conlcusion.
- The step size utilized for EBSD characterization should be specified. This is especially important here since the KAM map is used for analysis.
- Lots of typos and errors. For example, line 139 and 141, ‘which reducing’ and ‘which increasing’ should be ‘which reduces’ and ‘which increases’, respectively; line 159, ‘particles has’ should be ‘particles have’; line 166-167, the sentence starting with ‘The average…’ lacks proper verb; Fig. 14 caption, there seems to be some word(s) missing before ‘results’. There are many more throughout the paper. A thorough check of the paper is required.
Reviewer 2 Report
Effects of microstructural evolution on hardness of 7050 alloy
during processing
The authors have attempted to study the effect of microstructure and hardness on 7050 alloy during heat treatment/processing. The paper presents some results but I do have some concerns as given below:
1. The title is not appropriate. Hardness and microstructure evolution are a result of heat treatment/processing. So title has to be changed.
2. Introduction should have more references. It should explain how the authors' work is different from relevant literature.
3. The authors should explain the novelty of this work.
4. The materials and methods section should contain details about equipment, methods, experimental procedure only. The microstructural observations should be reported in results and discussion session.
5. Section 3.1 is explained pointwise. The authors can explain this in a paragraph.
6. It would be prudent to express the hardness values as whole numbers.
7. Fig. 3: Explain why there is drop in values for samples deformed with WQ.
8. Fig. 7: The micron markers in both SEM and TEM images are same. This is a blunder. Then how to the authors claim the precipitate size in nm?
9. Figs. 9, 10, 12, 13: should include inverse pole figure map in order to conclude which is the orientation. Each colour should be given a legend.
10. Do the authors note any texture development after deformation? If so, what kind of texture is developed?
11. Explain how Fig. 11 is obtained?
12. Fig. 14 quality should be improved and Fig heading should start in capitals. Also, it would be good if the authors represent the data using standard plotting softwares other than excel.
13. Section 3.4 Evolution of hardness should be supported with experimental evidences.
14.Conclusions: ‘Fast cooling after deformation promoted the occurrence of abnormal recrystallization during heat treatment’ – What is the cause of abnormal recrystallization?
15. The references are not in format and some more relevant references representing research from all parts of the world (not a specific country) should be added.
Reviewer 3 Report
The paper “Effects of microstructural evolution on hardness of 7050 alloy during processing” is devoted to the microstructural evolution, and its influence on the hardness of 7050 alloy during preheating, post-deformation cooling, and heat treatment. This topic is very important for manufacturers to obtain metal products with the required level of properties. The paper is well written and the text is clear and easy to read. However, there are some questions below:
1) In the Introduction, the authors list only the facts but do not provide their critical analysis of the literature. Moreover, only 5 out of 20 references (25%) are not older than 5 years. The Introduction part should be improved, including by using recent and relevant publications.
2) There are no references to Figures 1a and 1b in the text of the paper. There is no description of the explanations of the figures “a” and “b” in the title of Figure 1.
3) The values of the volume fraction of precipitates in Table 1 are very high. 65.06% and 86.49% are unreal values. There is an error in the calculations. For example,
eq. (4) ?????????=(?⁄4)??^3
in ref. 17 is written as Vplatelet = π/4TD^2.
4) Accordingly, further calculations and analysis in chapter 3.2.2 and conclusion 2 also raise doubts.
5) How long DWHT and DAHT samples were solution treated at the temperature of 474 C?
In general, the paper is interesting, but the Introduction and the volume fraction calculations of precipitates should be rewritten and recalculated.
Round 2
Reviewer 2 Report
Comment No 7 not addressed
Author Response
We are sorry that the response for comment 7 are not easy to find due to a typo error in the response document. The following is the original response:
Point 7: Fig. 3: Explain why there is drop in values for samples deformed with WQ.
Response 7: We are sorry that the explanation about lower hardness values in the DW samples was not clear in the original manuscript. Since plastic strain can induce a significant acceleration of the precipitate coarsening processes, the average size and volume fraction of matrix particles of the deformed materials increased significantly. Herein, it is believed that dynamic precipitation during plastic deformation reduces the degree of supersaturation, slowing down the nucleation rate of precipitates during cooling process, thereby reducing the hardness by 5.6%-10.8% from the preheated materials. In order to clarify this issue, the statistical analysis of the precipitates in samples deformed with WQ have been added in section 3.2.2 and discussed in detail in section 3.4 in the revised manuscript.
Once again, thank you very much for your comments and suggestions.
Reviewer 3 Report
The paper was improved and may be accepted for publication.
Author Response
Thank you and the review again for your help!